# Antibacterial Potential of Crude Extracts from *Cylindrospermum alatosporum* NR125682 and *Loriellopsis cavernicola* NR117881

**DOI:** 10.3390/microorganisms13010211

**Published:** 2025-01-19

**Authors:** Albert Olufemi Ikhane, Foluso Oluwagbemiga Osunsanmi, Rebamang Anthony Mosa, Andrew Rowland Opoku

**Affiliations:** 1Department of Biochemistry and Microbiology, University of Zululand, Richards Bay 3886, South Africa; opokua@unizulu.ac.za; 2Department of Biochemistry, Genetics and Microbiology, University of Pretoria, Hatfield 0002, South Africa; rebamang.mosa@up.ac.za

**Keywords:** cyanobacteria, efflux pump, antibiotics resistance, membrane damage, lactamase

## Abstract

The challenges of antimicrobial resistance (AMR) to human health have pushed for the discovery of a new antibiotics agent from natural products. Cyanobacteria are oxygen-producing photosynthetic prokaryotes found in a variety of water habitats. Secondary metabolites are produced by cyanobacteria to survive extreme environmental stress factors, including microbial competition. This study presents the antibacterial activity and mechanism of the crude extracts from *Cylindrospermum alatosporum* NR125682 (A) and *Loriellopsis cavernicola* NR117881 (B) isolated from freshwater. The cyanobacteria were identified through 16S rRNA sequencing. Crude extracts were sequentially prepared using hexane, dichloromethane, and ethanol consistently. The minimum inhibition concentration (MIC), minimum bactericidal concentration (MBC) using the CSLI microdilution test protocol, and crude extract potential to inhibit the growth of the tested clinical bacteria strains were evaluated. The mechanism of action of the extracts including membrane damage, efflux pump, β-lactamase activity, DNA degradation, and extract–drug interaction was investigated using standard procedures. The hexane extract of B performed the best with a MIC (0.7–1.41 mg/mL) and MBC (1.41–2.81 mg/mL) range. All the crude extracts inhibited efflux pump activity against the bacteria tested. However, the extracts poorly inhibited β-lactamase. The ethanol extract of B exhibited the most appreciable antibacterial activity. The dichloromethane extract of B showed the highest significant DNA degradation potential, when compared with other samples. The extracts exhibited synergism when combined with erythromycin against some test bacteria, indicating primary microbial activity through membrane interactions. Hence, this study demonstrates the significance of cyanobacteria for antibiotic development.

## 1. Introduction

Antimicrobial resistance (AMR) is increasingly becoming a silent pandemic [1]. There has been major reporting of the escalating failure of the last line of antimicrobials, raising AMR to the top of the priority problem list of major government health organisations. Championing the microbial resistance battle are bacteria, with rapid resistance to major antibiotic classes being the most problematic [2]. Consequently, this has led various research groups to the search for novel and highly effective antibiotics. Secondary metabolites from numerous lifeforms have been the go-to source, for decades since penicillin was first discovered, for the discovery of antimicrobials [3]. It is no surprise that numerous studies view secondary metabolites as a reservoir for antibiotic discovery [3,4,5]. Alternative approaches to this problem such as phage therapy and the development of antibacterial peptides are also interesting avenues to solving this silent pandemic [6].

Typically, antibiotics inhibit bacterial growth through protein synthesis inhibition, cell wall and membrane disruption, and folate synthesis inhibition among others. However, bacteria possess complex and rapidly evolving layers of defenses against numerous antibiotics and their derivatives. A mixture of efflux, target structure alteration, and enzymatic degradation mechanisms have been the major and most studied vehicle of resistance in bacteria [1]. Enzymatic degradation is a rather useful resistance mechanism by bacteria. The β-lactams are a class of antibiotics that attack peptidoglycan synthesis, making them an important class of antibiotic due to this selectivity [7]. However, β-lactamases, a specialized group of resistance enzymes that degrade β-lactam drugs effectively, have been rapidly expressed by resistant bacteria. They achieve this through catalyzing the hydrolysis of the lactam ring that makes up the active group in β-lactam drugs. The mode of catalysis can be through a serine- or zinc-assisted nucleophilic attack on the carboxyl carbon of the amide group in the lactam ring, in the active site of the enzymes [8]. Coupled with antibiotic-degrading enzymes are efflux protein (pumps). These pumps are protein complexes that recognize antibiotics and are effectively utilized to reduce cytoplasmic drug concentrations to sub-toxic levels. These complexes are often mutated transport proteins obtained through intrinsic chromosomal mutation or through plasmid uptake. The transport mechanisms of most pumps are either based on energy generation through ATP hydrolysis or powered through ionic concentration gradients (mostly H^+^, or Na^+^) dictated by proton motive forces (PMFs) with the substrates’ movement coupled to the ions (symporters) or opposite to the ions’ transport (antiporters) [9]. During translocation, the pump undergoes various conformational changes, which ensures the translocation of the drug from the cytoplasm into the surrounding medium. These pumps recognize a wide range of substrates, making them multidrug efflux pumps, and this has been reported in a diverse array of bacterial pathogens [10].

Efflux pumps have been a primary target for antibiotic development along with other cellular targets, such as the peptidoglycan through lactamases’ inhibition; efflux multidrug resistance has been a thorn in the path of newer drug developments due to the numerous substrates they can eject [11]. Synergistic therapies are often employed as a method of cellular multitargeting against pathogens with multiple resistance factors, and synergism between two classes of antibiotic can often improve antibiotic therapy; however, resistance is often reported primarily due to the antibiotics being based on traditional molecular scaffolds [12]. The paradigm shifts in antimicrobial development from known natural sources with an established mode of actions to less studied natural organisms have primarily been fostered due to the rise in multidrug resistance [13].

Cyanobacteria are among the promising sources for novel antibacterial metabolites that have been studied sparsely. They can synthesize a wide variety of phytochemicals, including antibacterial compounds [13]. Their phytochemical reservoirs have been attributed to their ability to survive in a diverse range of habitats, often outcompeting and dominating as the major microbial species in different niches. Antibacterial polyketides, alkaloids, peptides, terpenes, lipids, and polyphenols produced by cyanobacteria have been reported, with the common elucidated mechanisms of action being the inhibition of efflux pump, DNA replication interference, and cell membrane disruption [14]. Therefore, exploiting this potential in novel antibiotic development may be an avenue for the discovery of novel compounds and this study seeks this out.

We have previously demonstrated the antioxidant capacity and the in silico evaluations (with good binding affinity of −6.6, −6.3 Kcalmol^−1^) obtained following molecular docking against the β-lactamase of the extracts of *Cylindrospermum alatosporum* NR125682 *and Loriellopsis cavernicola* NR117881 [15]. However, there is scanty information on the antibacterial potential and mechanism of actions of the extract of freshwater cyanobacteria in the literature. Hence, this study reports the in vitro antibacterial activity and mechanism of action of crude extracts of *Cylindrospermum alatosporum* NR125682 and *Loriellopsis cavernicola* NR117881 isolated from a freshwater source. This study signifies an important step toward unveiling a new antibiotic remedy on the ongoing battle against antimicrobial resistance pathogens.

## 2. Materials and Methods

### 2.1. Chemical Reagents

All chemicals used were of analytical grade purchased from Sigma-Aldrich Company Limited (Steinheim, Germany). The BioTek Synergy HT plate reader (BioTek Instrument, Winooski, VT, USA) was used for all absorbance reading.

### 2.2. Isolation, Characterization, and Extraction

The details of the isolation, characterization, cultivation, and extraction of the cyanobacteria have been previously reported [15]. Briefly, the cyanobacteria were isolated from a freshwater source enriched with BG-11 medium and cultivated for 21 days under continuous illumination. A series of re-plating was carried out to isolate single and pure colonies that were identified using 16S rRNA. The cultivated cells were harvested through centrifugation and freeze drying, and sequentially extracted with hexane, dichloromethane, and 70% ethanol. The extracts were concentrated using a rotary evaporator and resuspended in 1% dimethyl sulfoxide.

### 2.3. Bacteria Strains

The bacteria (clinical bacteria strains, previously isolated from diabetic foot ulcers) utilized in this study were obtained from the Department of Biochemistry and Microbiology, University of Zululand. *Alcaligenes faecalis* CP033861, *Micrococcus luteus* KT805418, *Staphylococcus sciuri* MN788638, *Bacillus* sp. MH412683, *Glutamicibacter creatinolyticus* MT235529, *Corynebacterium striatum* MN121138, *Klebsiella aerogenes* CP035466, *Glutamicibacter creatinolyticus* CP034412, and *Staphylococcus aureus* AP025177 were used in this study. *Staphylococcus aureus* subsp. *aureus* ATCC25923 and *Bacillus subtilis* subsp. *spizizenii* ATCC6633 were used as control strains.

### 2.4. Antibacterial Susceptibility Testing

Susceptibility testing was performed on the bacteria listed in the bacteria strains section, following the Clinical and Laboratory Standard Institute (CLSI) guidelines to determine the Minimum Inhibitory Concentration (MIC) as well as Maximum Bactericidal Concentration. Erythromycin and penicillin were used as antibiotic control [16].

#### 2.4.1. Minimum Inhibitory Concentration (MIC)

The test extracts were prepared in Mueller–Hinton broths (MHBs). The stock extracts (100 µL) were added to 100 µL of MHB in the first well of the sterile 96-well microplates, with the only exceptions being the growth control wells. Two-fold serial dilutions were performed vertically on the 96-well microplates to create a concentration gradient of between 11.33 and 0.03 mg/mL, depending on the extract.

To prepare the inoculum, the test organisms were grown in MHB broth for a period of 24 hrs, after which the bacterial cultures were adjusted to a 0.5 McFarland standard by measuring the broth density absorbance at 625 nm. Afterwards, inoculation followed through the addition of 10 µL of the prepared inoculum for each bacterium into the microplate’s wells, including the control wells, bringing the final test bacterium concentration to 5.0 × 10^5^ CFU/mL. Each microplate was fitted with a plastic cover and incubated at 35 ± 2 °C for 30 h. To observe cell viability in the wells, 40 µL of 0.2 mg/mL Iodonitrotetrazolium chloride (INT) was added to the wells after the incubation period. An observable color change because of the reduction in INT was recorded. The MIC was regarded as the concentration where little to no color change was observed [16].

#### 2.4.2. Minimum Bactericidal Concentration (MBC)

The minimum bactericidal concentration was evaluated by subculturing from the MIC well and concentrations above the MIC into freshly prepared agar plates. No visible observable bacterial growth on the plates indicated bacterial unviability and thus a bactericidal effect of the extracts at the test concentration.

### 2.5. Lactate Dehydrogenase (LDH) Release Assay

Bacteria showing susceptibility to the extracts were grown and incubated with the MBC or four folds the MIC of the microalgal extracts, overnight. The microbial cultures were centrifuged (5000× *g*; 5 min). The supernatant (100 μL) was mixed with 100 μL of a lactic acid dehydrogenase substrate mixture (54 mM lactic acid, 0.28 mM of phenazinemethosulfate, 0.66 mM p-iodonitrotetrazolium violet, and 1.3 mM NAD^+^). The pyruvate-mediated conversion of 2,4-dinitrophenyl-hydrazine into a visible hydrazone precipitate was measured on the BioTek microplate reader at 492 nm. The loss of membrane integrity, due to the extract effect, was compared to the lysing of the cells treated with 3% Triton X-100. The cytotoxicity in the LDH release test was calculated using the following formula:% LDH Cytotoxicity = (E − C)/(T − C) × 100,
where E is the experimental absorbance of the cell cultures, C is the control absorbance of the cell medium, and T is the absorbance corresponding to the maximal (100%) LDH release of 3% Triton X-100 (positive control) lysed cells [17].

### 2.6. Rhodamine 6G Uptake (Efflux Pump Inhibition)

The activities of the cyanobacterial extracts on the efflux pump were tested for by their inhibition of Rhodamine 6G (R6G) cytoplasmic accumulation using the method described by Sewanu et al. [17]. Resistant bacteria were cultured overnight at 28 °C with shaking (110 rpm). After 24 h, cells were centrifuged at 4000× *g* for 5 min and washed twice with phosphate buffer saline (PBS, pH 7.2). Cells were centrifuged again and resuspended at 40 mg/mL in PBS containing 10 mM sodium azide (NaN_3_). R6G was added at a final concentration of 10 µM and cells were placed in an incubator (37 °C, 120 rpm) for 1 h. Cells were then divided into two portions, tube 1 and tube 2. Cells were centrifuged and washed as previously described. The cells in tube 1 were resuspended in PBS containing 1 M glucose, while the cells in tube 2 were resuspended in PBS alone. Cyanobacterial extracts (MBC concentration) were added to the cells containing glucose to make a final concentration of 100 µM. Both tubes were placed in an incubator with agitation for 30 min at 37 °C. The cells were then centrifuged and the supernatant discarded. The remaining pellet was resuspended in 0.1 M glycine (pH 3) and placed in the shaking incubator overnight. After 24 h, the cells were centrifuged for 10 min at 4000× *g,* the supernatant was collected, and the absorbance was read at 527 nm. The accumulation of the R6G was expressed as percentage accumulation in the cells. The percentage accumulation of R6G inside cells after exposure to glucose, the cyanobacterial extracts, and standards was calculated using the following formula: R6G % Accumulation = (1 − A_t_/A_o_) × 100
where A_t_ is the absorbance of the test compound, and A_o_ is the absorbance of the control in the presence of glucose only. Beberine was used as a standard.

### 2.7. β-Lactamase Inhibitory Activity

β-Lactamase inhibition assay was conducted as described by Yang, et al. [18], with some modifications. The MBC concentration of the extracts was incubated with β-lactamase (45 nM) for 15 min in 50 mM phosphate buffer (pH 7.0), containing 1 mg/mL BSA. Following the incubation, nitrocefin was added at a concentration of 1 mM, and then incubated for 30 min. The hydrolysis of nitrocefin was monitored by measuring the increase in the OD_486_ value at 5 min intervals. The slope value for each extract concentration as well as the control was then used to calculate the %inhibition using the following formula: (1 − S_t_/S_o_) × 100
where S_t_ is the slope value of the test and S_o_ is the slope value of the control.

The concentration of the extract that reduced enzyme activity by half was recorded as the IC_50_ value.

### 2.8. DNA Degradation in Bleomycin-Fe^3+^ DNA System

The extracts’ ability to degrade DNA was evaluated by preparing a mixture of DNA (0.1 mg/mL), MgCl_2_ (0.05 M), and FeCl_3_ (0.5 mM) in tris buffer (1 M, pH 7.4), into which the extracts were added and incubated at 37 °C for 30 min. Bleomycin was used as the standard. Afterwards, 1 mL of TBA (1%) and 1 mL of HCl (25%) were added and boiled for 10 min. The resulting solution was extracted into butanol and centrifuged (15 min, 5000× *g*). The top layer was pipetted into 96-well plates and the absorbance was read at A_532_.

The percentage degradation in relation to bleomycin was evaluated using the following formula:% Degradation = (E − C)/(B − C) × 100
where E is the test absorbance, C is the absorbance of the control with no bleomycin and test, and B is the absorbance of the bleomycin degradation system, as described by Burger et al. [19].

### 2.9. Extract–Drug Interactions

The interactions between the extracts and erythromycin were performed in 96-well microtiter plates, as described by Penduka et al. [20], using the chequerboard method. The starting antimicrobial combination (MIC concentration) was prepared in double strength Mueller–Hinton broth and serially diluted to make different test concentrations in the microtiter plate. Each well contained 100 μL of the test antimicrobial combination. A volume of 20 μL of the standardized 0.5 MacFarland test bacteria was added into the test wells. Sterility wells containing broth only and growth control wells containing the bacteria and broth only were also added in each microtiter plate. The MICs of the test combination were determined after 18–24 h of incubation at 37 °C, using the INT method mentioned in the MIC determination. The interactions were interpreted using Fractional Inhibitory Indices (FICs). The FIC indices of the extracts (FICEs) were calculated as the ratio of the MIC value of the extracts in combination over the MIC value of the extracts alone, and the FIC index of the antibiotic (FICA) was calculated as the ratio of the MIC value of the antibiotic in combination over the MIC value of the antibiotic alone. The overall FIC index (ΣFIC) was calculated as the summation of the FICE and the FICA. The interactions were interpreted as synergism when the ΣFIC index ≤ 0.5, additive when 0.5 < ΣFIC index ≤ 1, and indifference when 1 < ΣFIC index < 4, whilst antagonism was defined as when the ΣFIC index is ≥4. The test was performed in triplicates.

### 2.10. Data Analysis

Data are presented as the mean ± standard deviation (SD), *n* = 3. Statistical differences between the groups were performed by one-way analysis of variance (ANOVA), and statistically significant difference was considered at *p* < 0.05.

## 3. Results

### 3.1. Minimum Inhibitory Concentration (MIC)

The results of the MIC evaluation of the crude extracts are presented in Table 1. The ethanol extract of *L. cavernicola* demonstrated a significant (*p* < 0.05) broad range of activity against both Gram bacterial types utilized, with the lowest concentration observed recorded for this extract, notably against the tested Gram-positive bacteria. The MIC range for the ethanol extract (**BE** in Table 1) of *L. cavernicola* was between 0.7 and 1.41 mg/mL, 1.46 and 11.67 mg/mL for the dichloromethane extract of *L. cavernicola* (**BD**), and 3.34 mg/mL was the inhibitory threshold for the hexane extract of *L. cavernicola* (**BH**). On the other hand, the extracts of *C. alatosporum* demonstrated moderate activity, with MIC values ranging between 0.6 and 10.71 mg/mL for all extracts. However, it was observed that the lowest concentration was against *K. aerogenes* CP035466, *G. creatinolyticus* CP034412, and *S. aureus* AP025177 at 0.67 mg/mL for the ethanol extract of *C. alatosporum* (**AE**), and *A. faecalis* CP033861 and *S. aureus* AP025177 at 1.88 mg/mL for the dichloromethane extract of *C. alatosporum*. The hexane extract of *C. alatosporum* showed significant (*p* < 0.05) the lowest concentration against *S. aureus* ATCC 25923 and *B. subtilis* ATCC 6633. The bacteria were resistant to the standard antibiotics used in this study. This corresponds with the reported resistance genes in their BLAST databases, having been confirmed using 16S rRNA. Notably, both ethanol extracts performed better than the remaining two solvents.

### 3.2. Minimum Bactericidal Concentration (MBC)

The results obtained from the MBC evaluation of the extracts are presented in Table 2. The ethanol extract of *L. cavernicola* demonstrated a broad bactericidal range of activity against the positive and negative Gram bacterial types utilized. The BE showed significant (*p* < 0.05) lowest concentration when compared with other extracts and exhibited significant MBC against the bacteria utilized. In general, its MBC range fell between 1.41 and 2.81 mg/mL. The rest of the extracts showed partial bactericidal activity, in terms of the number of bacteria susceptible to bactericidal activity. For **BD,** the lowest MBC value of 2.92 mg/mL was demonstrated against *A. faecalis* CP033861, while **BH** recorded no MBC values against all tested bacteria. For **AE**, MBC values were observed in all tested bacteria except *A. faecalis* CP033861, *M. luteus* KT805418, *S. sciuri* MN788638, and *Bacillus* sp. MH412683. For **AD,** the only observable bactericidal activity was against *Alcaligenes faecalis* CP033861, and **AH** demonstrated MBC values against *G. creatinolyticus* MT235529, *G. creatinolyticus* CP034412, *S. aureus* AP025177, and *S. aureus* ATCC 25923. However, it is apparent that the ethanol extract of *L. cavernicola* exhibited the most significant (*p* < 0.05) antibacterial activity. This extract, unless stated otherwise, was therefore used in subsequent studies.

### 3.3. Lactate Dehydrogenase (LDH) Release Assay (Membrane Damage)

The potential of the ethanol extract of *L. cavernicola* to induce membrane damage was investigated. The results (Table 3) point to the potency of the extract against the tested bacteria of the Staphylococcus genus. Of the nine bacteria tested, the extract exhibited no activity against *C. striatum* MN121138, *K. aerogenes* CP035466, and *G. creatinolyticus* CP034412, although the observed activity on *Alcaligenes faecalis* suggests the potential of the extract to circumvent the outer membrane of Gram-negative bacteria. However, the extracts displayed strong significant (*p* < 0.05) lactate dehydrogenase release in both *Staphylococcus* bacteria.

### 3.4. Rhodamine 6G Uptake (Efflux Pump Inhibition)

The ethanol extract of *L. cavernicola* was screened for its ability to inhibit efflux pumps in the presence of glucose (Figure 1). The crude extract showed a significant accumulation of R6G dye in most of the tested bacteria (especially in *M. luteus*, *Bacillus* sp., *C. striatum,* and *G. creatinolyticus)*. Hence, this indicates the efficiency of the extract as efflux pump inhibitors. Unlike the lactate dehydrogenase assay, we observed R6G accumulation in *C. striatum* MN121138, *K. aerogenes* CP035466, and *G. creatinolyticus* CP034412; this further indicates membrane interactions being the primary inhibitory mode of the active component in the extract.

### 3.5. β-Lactamase Inhibition Activity

To investigate the β-lactamase inhibitory potential of the extracts, we utilized all of the extracts. Except for the hexane extract of *C. alatosporum* with a measurable IC_50_ of 5.6 mg/mL, all the other extracts were poor inhibitors of β-Lactamase activity (Figure 2).

### 3.6. DNA Degradation in Bleomycin-Fe^3+^ DNA System

All the extracts were evaluated for their effect on DNA. Bleomycin is an antibiotic with an ability to effect Fe^3+^ oxidative damage of DNA molecules, irreversibly forming a malondialdehyde-like complex. The percentage cleavage of DNA by the extracts, in comparison to the bleomycin complex, is displayed in Figure 3. The results indicated the two-fold cleavage potential of the DCM extract of *L. cavernicola* (BD).

### 3.7. Extract–Erythromycin Interactions

Using the *ΣFIC* index, we evaluated the ethanol extract of *L. cavernicola* for its potential to restore antibacterial activity to erythromycin for some of the bacteria previously tested. We hypothesized that the inhibition of efflux pump previously observed may improve erythromycin activity in vitro since one of the major resistance mechanisms reported for this antibiotic is through efflux pumps. The results of the interactions of the extract with erythromycin are shown in Table 4. Synergism was observed in all but *A. faecalis* and *G. creatinolyticus,* with both being indifferent.

## 4. Discussion

The burden of antimicrobial resistance has become an ever-present threat in human medicine [21]. Indeed, the rise in multidrug resistance bacteria coupled with the marked reduction in the development of novel antibiotics has resulted in an era where simple bacterial infections are increasingly becoming difficult to treat [22,23]. There is a need for the development of new antimicrobials, especially newer classes of effective antibiotics, alternative to the current antibiotic classes, for which resistance has not yet developed [23]. Therefore, there is growing research into other sources, such as cyanobacteria and microalgae, for the potential development of new antibiotics, primarily because they possess a rich consortium of secondary metabolites.

There have been a series of studies on cyanobacteria that has shed light on this bacterial phylum as a promising source for the discovery of a myriad of different metabolites, including antimicrobials [13,14,24,25]. Cepas, et al. [26] reported the antibacterial effect of a series of cyanobacteria extracts, with antibiofilm activity. Shishido, et al. [27] reported the antimicrobial activity of *Nostoc* sp. and *Fischerella* sp. extracts, further establishing cyanobacteria as a source for antibacterial products.

In similar fashion, the activity of the extracts of *C. alatosporum NR125682* and *L. cavernicola NR117881* observed in this study (Table 1) reveals a pattern with bacterial growth inhibition against both resistant Gram-negative and -positive bacteria pointing to the antibiotic potential of the previously reported constituents [15] in the extracts of the two organisms studied. Surprisingly, the GC-MS data revealed only two compounds for the ethanol extract of *L. cavernicola,* pointing to the possibility of other uncharacterized metabolites. Regarding the two observed compounds, to the best of our knowledge, there has not been a detailed report of the antibacterial properties of (1) 7,9-Di-tert-butyl-1-oxaspiro (4,5) deca-6,9-diene-2,8; however, there have been reports of its potential toxicity leaching into water bodies from polyethylene pipes [28], and it has also been reported in extracts of different organisms, including plants, with antibacterial/antifungal activity [29]. On the other hand (2), nonanal has been extensively discussed for its antibacterial and antifungal activity [30]. Thus, the observed antibacterial activity may be attributed to the presence of both compounds.

Cell membranes are an important part of the cellular structure of bacteria; they constitute an additional effective barrier between the cytoplasm and extracellular structures of bacteria [31]. Contained in the membrane are proteins significant for the physiology of the bacterial cell, like the transport of essential materials and nutrients into the cell as well as the movement of toxic material out from the cell [32]. The breakdown of this critical structure leads to microbial death, which makes it a major target in antibiotic development. Antibiotics such as Daptomycin, Polymyxins, and Vancomycin target cell membranes in their mode of action [33,34,35]. Membrane disruption has previously been reported as a destruction mechanism of cyanobacteria extracts [13]. The high percentage release of LDH observed (Table 2) in both Staphylococcus bacteria indicates the ability of the extract to damage Gram-positive membranes. The notoriety of *Staphylococci* resistance is well established [36], especially for methicillin resistance; therefore the ability of the extract to effectively damage the membrane of this bacterial genus further elucidates the antibiotic potential of the cyanobacterial extracts. The extracts were also able to damage the membrane of *A. faecalis,* suggesting a broad spectrum of activity against both types of Gram bacteria. The extract effectiveness on *A. faecalis* suggests a capability to infiltrate Gram-negative double membrane defense with probable interactions with the outer and inner membrane. At first glance, better activity was observed on bacteria with a spherical/circular morphology, as compared with rod-shaped morphology [37]; this is, however, speculative, and would require further research.

Critical to this study is the establishment of potential resistance mechanism inhibitions, as this is the core of this research. Despite molecular docking results [15] predicting a strong binding affinity to the enzyme, the extracts, unfortunately, were poor inhibitors of β-lactamase (Figure 2), and the IC_50_ value obtained for the most active hexane sample of *C. alatosporum* was not of clinical significance. However, since the crude extracts were tested and not the pure compound (used in the in silico studies) itself, this suggests that the poor activity may be a consequence of the interference of other compounds contained in the extracts, or that the compound exists in the extract at a very low (ineffective) concentration.

Cyanobacteria have been reported to possess efflux pump-inhibiting compounds [26,38,39]. The ethanol extract of *L. cavernicola* was able to promote the intracellular accumulation of R6G, indicating efflux inhibition. Over 50% accumulation of the dye was observed in all the bacteria tested (Figure 1). It is noted that the extract exhibited the potential to inhibit the efflux pumps of *S. aureus* (well documented to carry multiple multidrug efflux systems [40]) and *A. faecalis* (another pathogen in nosocomial and opportunistic infections [41]). Efflux pumps have been implicated in the proliferations and successes of biofilms, cellular toxin reduction, evasion of immune cells, and quorum sensing [42,43]; a major inhibition process is attributed to the competitive binding of the inhibitor to the efflux system, displacing the actual substrate at various levels of the protein structure or through the disruption of the efflux’s proton motive system, and for ATPase-bound efflux systems, the inhibition of the energy-generating enzyme may also reduce efflux effectivity [44]. Erythromycin resistance is facilitated by efflux systems [45], and the synergism observed (Table 3) in combination with erythromycin suggests that the ethanol crude extract’s membrane disruption and efflux inhibition can promote the activity of erythromycin by reducing the amount of antibiotic pumped out of the bacteria cell, thereby boosting its activity. No antagonism was observed. Bleomycin is an antibiotic with the ability to affect a Fe^3+^ oxidative damage of DNA molecules, irreversibly forming a malondialdehyde-like complex. We observed a similar activity of the DCM extract of *L. cavernicola* compared with bleomycin, suggesting a compound(s) with the ability to bind DNA and make it venerable for degradation. Cyanobacteria have been reported to produce toxins capable of enacting DNA damage [46]. The rest of the extracts, however, displayed a poor DNA binding effect in the system utilized. Overall indications of promising antibacterial activity are apparent in the extracts of *L. cavernicola*, particularly pointing to bacterial inhibition through membrane disruption and efflux inhibition.

## 5. Conclusions

As the search for an alternative antibacterial intensifies, in this study we contribute a preliminary screening of the potential of two cyanobacteria as a source for novel antibiotics. The ethanol extracts appear to display the highest antibacterial potency, and this infers the presence of antibacterial metabolites. It is, however, apparent that the major antibacterial mechanism is through membrane interactions and the inhibition of efflux pumps. This indicates the potential of the extracts (especially ***Loriellopsis cavernicola NR117881***) for novel antibiotic development targeting critical inhibition areas as well as a capacity for antibiotic adjuvant development. However, not reported in this study is the cytotoxicity of the extracts; we therefore recommend this for further studies. Additionally, we advise for the isolation and elucidation of bioactive compounds within the crude extracts; we predict they are potential candidates as novel antibacterial compounds.

## Figures and Tables

**Figure 1 microorganisms-13-00211-f001:**
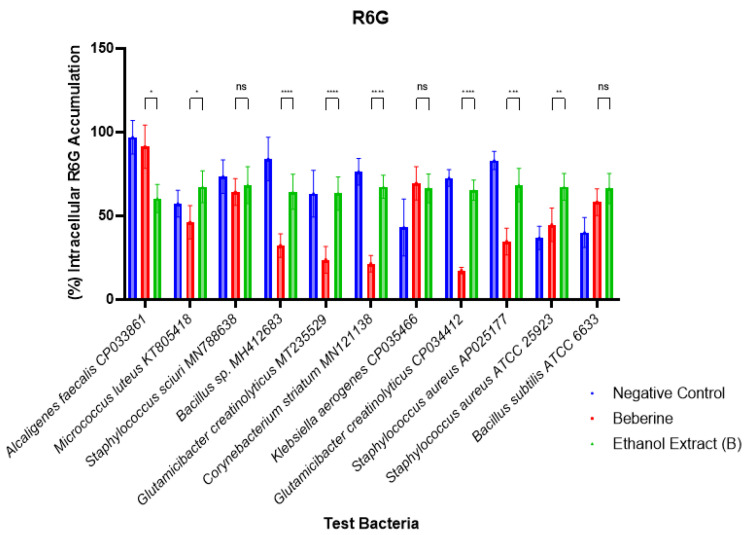
Effect of the ethanol extract of Loriellopsis cavernicola on intracellular accumulation of Rodamine 6G dye. *p* value ns: no significance, * < 0.05, ** < 0.005, *** < 0.0005, **** < 0.0001.

**Figure 2 microorganisms-13-00211-f002:**
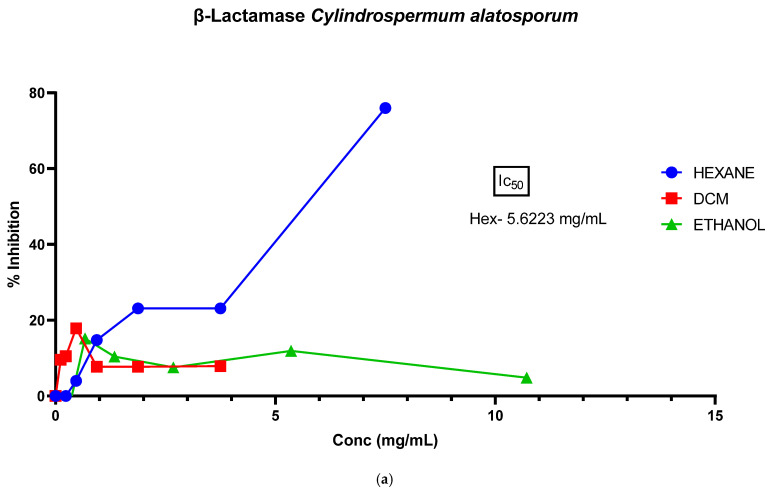
Lactamase inhibitory capacity of the crude extracts. (**a**) Inhibitory capacity of *Cylindrospermum alatosporum* crude extracts; (**b**) inhibitory capacity of *Loriellopsis cavernicola* crude extracts.

**Figure 3 microorganisms-13-00211-f003:**
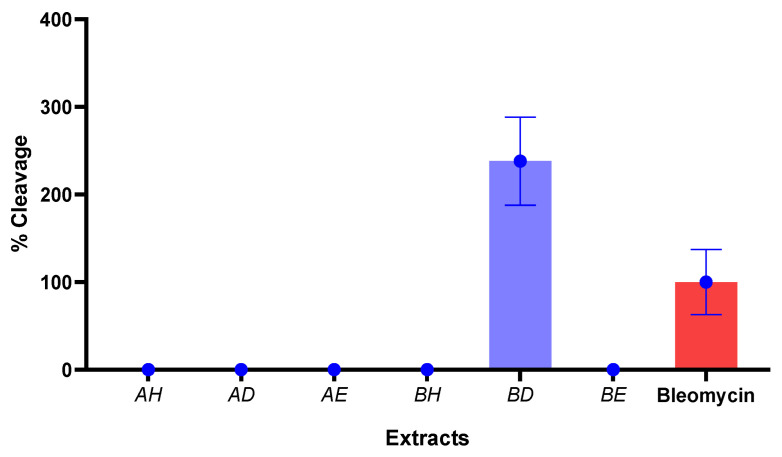
Percentage cleavage of the crude extracts (AH: Hexane, *Cylindrospermum alatosporum* NR125682; AD: Dichloromethane, *Cylindrospermum alatosporum* NR125682; AE: Ethanol, *Cylindrospermum alatosporum* NR125682; BH: Hexane, *Loriellopsis cavernicola* NR117881; BD: Dichloromethane, *Loriellopsis cavernicola* NR117881; BE: Ethanol, *Loriellopsis cavernicola* NR117881).

**Table 1 microorganisms-13-00211-t001:** Minimum inhibitory concentration (mg/mL) of the crude extracts (AH: Hexane, *Cylindrospermum alatosporum* NR125682; AD: Dichloromethane, *Cylindrospermum alatosporum* NR125682; AE: Ethanol, *Cylindrospermum alatosporum* NR125682; BH: Hexane, *Loriellopsis cavernicola* NR117881; BD: Dichloromethane, *Loriellopsis cavernicola* NR117881; BE: Ethanol, *Loriellopsis cavernicola* NR117881; ERY: Erythromycin; PEN-G: Penicillin G). Different alphabet letters indicated significant differences, (-) not determined.

	AH	AD	AE	BH	BD	BE	ERY	PEN-G
*Alcaligenes faecalis* CP033861	7.50 ± 0.01 ^b^	1.88 ± 0.41 ^c^	10.72 ± 0.95 ^a^	3.34 ± 0.31 ^d^	1.46 ± 0.05 ^f^	1.41 ± 0.003 ^f^	1.00 ± 0.10 ^f^	1.00 ± 0.10 ^f^
*Micrococcus luteus* KT805418	7.50 ± 0.02 ^b^	3.75 ± 0.51 ^c^	5.36 ± 0.4 ^a^	3.34 ± 0.32 ^c^	2.92 ± 0.08 ^h^	1.41 ± 0.00 f	1.00 ± 0.52 ^f^	1.00 ± 0.21 ^f^
*Staphylococcus sciuri* MN788638	7.50 ± 0.01 ^b^	3.75 ± 0.63 ^c^	5.36 ± 0.02 ^a^	3.34 ± 0.19 ^c^	5.83 ± 0.91 ^a^	0.70 ± 0.002 ^f^	1.00 ± 0.01 ^f^	-
*Bacillus* sp. MH412683	3.75 ± 0.08 ^c^	3.75 ± 0.51 ^c^	5.36 ± 0.45 ^a^	3.34 ± 0.14 ^c^	5.83 ± 0.96 ^a^	0.70 ± 0.11 ^f^	1.00 ± 0.09 ^f^	-
*Glutamicibacter creatinolyticus* MT235529	3.75 ± 0.02 ^c^	3.75 ± 0.21 ^c^	5.36 ± 0.56 ^a^	3.34 ± 0.27 ^c^	11.67 ± 1.8 ^a^	0.70 ± 0.23 ^f^	0.50 ± 0.01 ^f^	1.00 ± 0.01 ^f^
*Corynebacterium striatum* MN121138	7.50 ± 0.15 ^b^	3.75 ± 0.16 ^c^	2.68 ± 0.15 ^c^	3.34 ± 0.24 ^c^	11.67 ± 2.6 ^a^	1.41 ± 0.05 ^f^	-	-
*Klebsiella aerogenes* CP035466	7.50 ± 0.01 ^b^	3.75 ± 0.01 ^c^	0.67 ± 0.03 ^f^	3.34 ± 0.61 ^c^	11.67 ± 0.1 ^a^	0.70 ± 0.1 ^f^	1.00 ± 0.05 ^f^	-
*Glutamicibacter creatinolyticus* CP034412	3.75 ± 0.05 ^c^	3.75 ± 0.15 ^c^	0.67 ± 0.04 ^f^	3.34 ± 0.01 ^c^	11.6 ± 0.90 ^a^	0.70 ± 0.2 ^f^	-	-
*Staphylococcus aureus* AP025177	1.88 ± 0.09 ^b^	1.88 ± 0.1 ^f^	0.67 ± 0.01 ^f^	3.34 ± 0.05 ^f^	11.6 ± 1.89 ^a^	0.70 ± 0.1 ^f^	0.06 ± 0.01 ^f^	-
*Staphylococcus aureus* ATCC 25923	0.94 ± 0.01 ^b^	3.75 ± 0.21 ^c^	1.34 ± 0.31 ^f^	3.34 ± 0.21 ^f^	11.67 ± 1.15 ^a^	0.70 ± 0.014 ^f^	0.06 ± 0.01 ^f^	0.06 ± 0.08 ^f^
*Bacillus subtilis* ATCC 6633	0.94 ± 0.02 ^b^	3.75 ± 0.02 ^c^	1.34 ± 0.11 ^f^	3.34 ± 0.01 ^c^	11.67 ± 1.50 ^a^	0.70 ± 0.003 ^f^	0.06 ± 0.01 ^f^	-

**Table 2 microorganisms-13-00211-t002:** Minimum bactericidal concentration (mg/mL) of the crude extracts (AH: Hexane, *Cylindrospermum alatosporum* NR125682; AD: Dichloromethane, *Cylindrospermum alatosporum* NR125682; AE: Ethanol, *Cylindrospermum alatosporum* NR125682; BH: Hexane, *Loriellopsis cavernicola* NR117881; BD: Dichloromethane, *Loriellopsis cavernicola* NR117881; BE: Ethanol, *Loriellopsis cavernicola* NR117881; ERY: Erythromycin; PEN-G: Penicillin G). Different alphabet letters indicate significant differences, (-) not determined.

	AH	AD	AE	BH	BD	BE	ERY	PEN-G
*Alcaligenes faecalis* CP033861	-	3.75 ± 0.11 ^b^	-	-	2.92 ± 0.01 ^a^	2.81 ± 0.01 ^a^	-	-
*Micrococcus luteus* KT805418	-	-	-	-	5.83 ± 0.01 ^b^	2.81 ± 0.14 ^a^	-	-
*Staphylococcus sciuri* MN788638	-	-	-	-	11.67 ± 0.01 ^b^	1.41 ± 0.02 ^a^	-	-
*Bacillus* sp. MH412683	-	-	-	-	11.67 ± 0.01 ^b^	1.41 ± 0.15 ^a^	-	-
*Glutamicibacter creatinolyticus* MT235529	7.50 ± 0.5 ^c^	-	10.72 ± 1.2 ^b^	-	-	1.41 ± 0.12 ^a^	-	-
*Corynebacterium striatum* MN121138	-	-	5.36 ± 0.3 ^b^	-	-	2.81 ± 0.15 ^a^	-	-
*Klebsiella aerogenes* CP035466	-	-	2.68 ± 0.1 ^a^	-	-	1.41 ± 0.23 ^a^	-	-
*Glutamicibacter creatinolyticus* CP034412	7.50 ± 0.014 ^c^	-	2.68 ± 0.19 ^a^	-	-	1.41 ± 0.14 ^a^	-	-
*Staphylococcus aureus* AP025177	7.50 ± 0.7 ^c^	-	2.68 ± 0.14 ^a^	-	-	1.41 ± 0.08 ^a^	-	-
*Staphylococcus aureus* ATCC 25923	3.75 ± 0.021 ^c^	-	2.68 ± 0.15 ^a^	-	-	1.41 ± 0.11 ^a^	-	-
*Bacillus subtilis* ATCC 6633	-	-	2.68 ± 0.01 ^a^	-	-	1.41 ± 0.31 ^a^	-	-

**Table 3 microorganisms-13-00211-t003:** Membrane damage activity against the tested bacteria. Different alphabet letters indicate significant differences.

Bacteria	%Enzyme Release	%Enzyme Release in Relation (Where 1 Indicates Equal 100% Activity as 3% Triton X-100)
*Alcaligenes faecalis* CP033861	117 ± 0.01 ^c^	1.0
*Micrococcus luteus* KT805418	89 ± 0.02 ^c^	0.9
*Staphylococcus sciuri* MN788638	600 ± 0.01 ^f^	6
*Bacillus* sp. MH412683	39 ± 0.03 ^c^	0.4
*Glutamicibacter creatinolyticus* MT235529	32 ± 0.04 ^c^	0.3
*Corynebacterium striatum* MN121138	0	0
*Klebsiella aerogenes* CP035466	0	0
*Glutamicibacter creatinolyticus* CP034412	0	0
*Staphylococcus aureus* AP025177	640 ± 0.02 ^f^	6

**Table 4 microorganisms-13-00211-t004:** Erythromycin extract synergistic interactions.

Organism	FIC Index of *L. cavernicola* Extract	FIC Index of Erythromycin	ΣFIC	Interactions
*Bacillus* sp. MH412683	0.1	0.3	0.4	Synergism
*Glutamicibacter creatinolyticus* MT235529	0.1	0.3	0.4	Synergism
*Alcaligenes faecalis* CP033861	0.2	1.3	1.5	Indifference
*Staphylococcus aureus* AP025177	0.3	0.2	0.5	Synergism
*Glutamicibacter creatinolyticus* CP034412	0.5	0.3	0.8	Indifference

FIC: Fractional Inhibitory Concentration.

## Data Availability

The original contributions presented in this study are included in the article. Further inquiries can be directed to the corresponding authors.

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
