# Peer review of "Antibacterial Potential of Crude Extracts from Cylindrospermum alatosporum NR125682 and Loriellopsis cavernicola NR117881"

_microorganisms, 2025, doi:10.3390/microorganisms13010211_

Round 1

Reviewer 1 Report (Previous Reviewer 2)

Comments and Suggestions for Authors

microorganisms-3442465-peer-review-v1

Authors have improved the quality of the manuscript. However, the style of formatting is not satisfactory, and authors show clear negligence regarding this point.

Ln93: Starin identification "NR125682" and etc. do not need to be in italics. Please, check and correct the entire manuscript.

After formally introducing the full Latin name of specific organism. in following occasions name need to be abbreviated according to the recommendations. Please, check the entire manuscript.

For all material and equipment used for the performance of the current research project, authors will need to provide information about the supplier. This information need to contain the name of the company and appropriate address, including the name of the city, state (in case of federal countries) in abbreviated way, and name of the country. Please, address of the headquarters of the company needs to be provided; Please, avoid providing names/address of the local distributors.

Ln123: subsp. do not need to be in italics. Please, check entire manuscript for similar adjustments

From Ln245: Several of mentioned microbial species were already introduced, please, abbreviate them according to instructions.

Ln246 and etc: C alatosporum need to be C. alatosporum. Please, check for similar adjustments.

Ln285: Staphylococcus need to be in italics.

Authors need to provide better quality figures.

Authors have improved the discussion section, and in current form is better presented.

Reference still need to be adjusted regarding the style. Well, some of the journal names are abbreviated, some are fully written. Some of the references are with provided doi, some not. All words in the journal names need to start with capital letters. Some of the reference titles are with each word with first capital letters, others not. Missing italics in some of the reference. All this shows kind on negligence from the authors regarding formatting process.

Author Response

Reviewer 2 Report (Previous Reviewer 3)

Comments and Suggestions for Authors

The authors have corrected the original version of their manuscript. It’s nice that in its present form the Manuscript corresponds to the journal Microorganisms and requires only minor revision.

Line 20. In the abstract, the authors designate the test as "maximum bactericidal concentration (MBC)", in section 2.4.2. Materials and Methods, the same test is designated as "Minimum Bactericidal Concentration (MBC)". Where is the right option?

Line 155 "5000g" - add a space between the value and the units.

The same, the line 182 “527nm”, line 201 “(1M…”

Line 199. “Dna” again, should it be DNA?

Line 243, 248 «C alatosporum» requires a dot after the abbreviation "C."

Same as line 262 for "L cavernicola"

Line 284. The authors provide a link to the data presented in Table 2 (Table 2), however, the actual data on inhibition are presented in Table 3.

Line 297. In "C. Striatum", "striatum" should not begin with a capital letter, but with a capital letter.

Figure 2a, the name of the ordinate axis is unreadable. Please correct the quality.

Line 345. It says: "All the extracts were evaluated or their effect on DNA" perhaps the authors meant "All extracts were evaluated for their effect on DNA"

Line 360-361. The authors refer to Table 3, "The results of the interactions of the extract with erythromycin are shown in Table 3." In fact, the results are presented in Table 4.

Table 4 "L. cavernicola" should be italicized

Line 427 "S. aureus" " should be italicized

Author Response

This manuscript is a resubmission of an earlier submission. The following is a list of the peer review reports and author responses from that submission.

Round 1

Reviewer 1 Report

Comments and Suggestions for Authors

The manuscript submitted by Ikhane et al. describes the antibacterial activity and mechanism of the crude extracts from Cylindrospermum alatosporum NR125682 and Loriellopsis cavenicola NR117881 isolated from freshwater. The The extracts exhibited synergism when combined with erythromycin against some test bacteria indicating primary microbial activity through membrane interactions. The manuscript is very interesting and well-written. However, I have some comments:

1) Please describe in more detail about the extraction techniques and the extract yield. Did the authors analyzed the bioactive substances by GC chromatography or NMR?

2) The introduction needs to cite more recent references. Some paragraphs need references. Please see the  attached file

3) The discussion needs to be re-written to discuss all the obtained results, not only focus on the mechanistic effects.

4) Bacterial species should be written in italic, please see the attached file

5) Other minor comments are attached in the attached file

Comments on the Quality of English Language

The manuscript is very interesting and well-written. 

Reviewer 2 Report

Comments and Suggestions for Authors

microorganisms-3332727-peer-review-v1

The paper present interesting results, however, need an extensive revision in order to be suggested for publication.

Authors need to find a balance between description of the applied methods. Some are presented with good levels of sufficient details, and then other with a very short levels of details.

In the paper is quite discordance between described information in material and methods and later presented results. Authors will need to correct the paper and maybe look for help from more experience colleagues to assist them in the structuring and formatting the manuscript.

Discussion is very basic and need an extensive upgrade and update. Authors need to discuss all the results presented in the manuscript.

Concussion is very generic.

Some of the references are not into the style. Examples Reference 4, 5 - name of the journal. Moreover, some of the journal names are abbreviated and other not... Please, check and correct. Please, check and see where italics need to be added for the references.

Ln 186: Please, provide more details

cavernicola? Please, check the spelling

Ln 39: Maybe will be more appropriate instead to antibiotics to say antimicrobials.

Ln 51: Please, in this and following occasions, use Greek symbol for beta.

Ln 111: Please, correct the text, 16S rRNA is for identification.

Ln 111 - 114: This section need a refernece.

Ln118-123: The strain identifications do not need to be in italics; You do not have identification of Bacillus MH412683? Moreover, Sp, do not need to be with capital S.

Strain ATCC 25923 need to be presented with full name: Staphylococcus aureus subsp. aureus; ATCC 6633 is Bacillus subtilis subsp. spizizenii

Ln125-128: Selection of the antibiotic needs to be according to recommendations from EFSA.

L145: any reference?

Ln167: reference need to be cited according to the instructions.

Ln207: Described by.... provide name....[18]

Table 1: The strain identifications do not need to be in italics. Please, correct

Ln248: Provide strain identifications

Ln251: What Table 1 again? Moreover, italics need to be added for the microbial names.

Ln273: Please, describe in Material and methods in the way how was described on Ln273.

Table 2 (Page 9), needs to be with different number? Add italics.

Reviewer 3 Report

Comments and Suggestions for Authors

The work of Albert Olufemi Ihane and co-authors "Antibacterial potential of crude extracts from Cylindrospemum alatosporum NR125682 and Loriellopsis cavenicola NR117881" is devoted to the study of the effectiveness of cyanobacteria extracts against pathogenic microorganisms. The authors conducted a series of tests and showed that cyanobacteria extracts exhibit antimicrobial activity in some cases. The work presents a number of interesting data.  The presented material corresponds to the profile of the journal Microorganisms.

Nevertheless, the work requires some revision. First of all, there is a question of setting up experiments related to conducting controls. In their work, the authors use hexane, dichloromethane and ethanol to obtain cyanobacteria extracts. This is understandable. Next, the extracts are evaporated and the precipitate is dissolved in dimethyl sulfoxide. Taking into account the fact that when conducting the reaction in 96-well microplates, 100 µl is first diluted to 100 µl of the medium, the concentration of DMSO in the first well is 50%. There is no data anywhere showing the effect of pure DMSO on cells in the same concentrations as (extract+ DMSO). The question is, has such control been carried out?

Section 2.8 "Rhodamine 6G Uptake (Efflux Pump Inhibition)", line 170. It is written that the authors used sodium azide. It is known that sodium azide is an antimicrobial agent and prevents the development of bacteria. The question is why azide was added in the middle of the experiment. And why, after separation into 2 test tubes, the cells were simply deposited, and not washed from azide for subsequent expression?

In addition, the authors should correct a number of inaccuracies in the presentation of the material.

 The authors should check the correctness of the name of the microorganisms in the title of the article. Cylindrospemum alatosporum or Cylindrospermum alatosporum?

Line 17. Please remove the sentence "The cyanobacteria were identified and characterized through 16S rRNA sequencing" from the abstract, since this information refers to your previous article, and not to the results presented. In addition, this phrase is very unfortunate in fact. What characterization, other than identification, is carried out by the 16SrRNA gene?

Line 18 “Crude extracts were prepared using hexane, dichloromethane and ethanol respectively.” Respectively to what? maybe a more accurate expression is not "respectively", but "consistently"? If it is implied that the extraction was carried out first with hexane, then with dichloromethane, and then with ethanol.

Line 26 “The Dichloromethane extract…” there is no need to start “Dichloromethane” with the capital letter.

Line 119 and 122 “Sp” and “Subtilis” should not be started with capital letter

Line 194 “Dna” should be DNA

All Latin names of bacteria, including those in the tables, in figures and in figures’ legends, should be italicized. And the index after the species name should not be italicized

Round 2

Reviewer 1 Report

Comments and Suggestions for Authors

I would like to thank the authors for their revision; the manuscript has greatly improved. Most of the previous comments have been addressed. However, the authors ignored some. I still have some comments:

1.        The result section needs to be written in more detail. Simply stating that results are shown in the table or figure is not informative enough.

2.        Tables and figures should be stand-alone; please expand the full names of all abbreviations in the legends

3.        —Again, the discussion needs to be improved. The introduction is too weak; „There is a rising need for the development of new antimicrobials!!!. Please highlight the problem of MDR and the need for alternatives, using more references. Please discuss the antibacterial effects in relation to the bioactive substances in the Cyanobacteria in more detail.? And refer to more references.

     Other minor revisions

1.        Please ensure that the suppliers of all materials used have been provided. i.e., Barberine?

2.        Please use a uniform style in writting 37°C or 37 °C;

1-       Table 1, please expand the full names of MIC and MBC.

2-       Table 2: %Enzyme release....  Not Enzyme Release

3-       Figure 1: (%) Intracellular R6G accumulation…..  Not Accumulation; Also, please expand the full name of R6G in the legend.

4-       Figure 2: Y axis in Figure a is not clear

5-       Table 3: please expand the full name of FIC in the legend.

Comments on the Quality of English Language

The English could be improved to more clearly express the research.

Reviewer 2 Report

Comments and Suggestions for Authors

microorganisms-3332727-peer-review-v1

Authors have performed only some editorials adjustments and in fact do not rally correct the paper. Most of the comments/suggestions from the original review were not answered. Even if the authors do not agree with the comments, then appropriate arguments needs to be provided. The paper present interesting results, however, need an extensive revision in order to be suggested for publication.

Authors need to find a balance between description of the applied methods. Some are presented with good levels of sufficient details, and then other with a very short levels of details.

In the paper is quite discordance between described information in material and methods and later presented results. Authors will need to correct the paper and maybe look for help from more experience colleagues to assist them in the structuring and formatting the manuscript.

Discussion is very basic and need an extensive upgrade and update. Authors need to discuss all the results presented in the manuscript.

Concussion is very generic.

Some of the references are not into the style. Examples Reference 4, 5 - name of the journal. Moreover, some of the journal names are abbreviated and other not... Please, check and correct. Please, check and see where italics need to be added for the references.

Ln 186: Please, provide more details

Ln 39: Maybe will be more appropriate instead to antibiotics to say antimicrobials.

Ln 111 - 114: This section need a refernece.

Ln118-123: You do not have identification of Bacillus MH412683? When working with bacillus, full identification is very important, since some species from Genus Bacillus are serious pathogens.

Ln123: subsp. do not need to be in italics

Ln125-128: Selection of the antibiotic needs to be according to recommendations from EFSA.

L145: any reference?

Ln167: reference need to be cited according to the instructions.

Ln207: Described by.... provide name....[18]

Table 1: The strain identifications do not need to be in italics. Please, correct

Ln248: Provide strain identifications

Ln251: What Table 1 again? Moreover, italics need to be added for the microbial names.

Ln273: Please, describe in Material and methods in the way how was described on Ln273.

Reviewer 3 Report

Comments and Suggestions for Authors

Dear authors, I cannot understand why you do not correct the error in the Latin name in the Title of your article. Look at the first name, not second! the letter 'R' in the word "Cylindrospemum" is missing.

Line 201. Please, check "Dna" or DNA?
